# Comparison of the Effect of Pan-Retinal Photocoagulation and Intravitreal Conbercept Treatment on the Change of Retinal Vessel Density Monitored by Optical Coherence Tomography Angiography in Patients with Proliferative Diabetic Retinopathy

**DOI:** 10.3390/jcm10194484

**Published:** 2021-09-29

**Authors:** Hongkun Zhao, Minzhong Yu, Lijun Zhou, Cong Li, Lin Lu, Chenjin Jin

**Affiliations:** 1State Key Laboratory of Ophthalmology, Zhongshan Ophthalmic Center, Sun Yat-Sen University, 54 South Xianlie Road, Guangzhou 510060, China; zhaohk5@mail2.sysu.edu.cn (H.Z.); zhljun@mail3.sysu.edu.cn (L.Z.); licong35@mail2.sysu.edu.cn (C.L.); lvlin@mail.sysu.edu.cn (L.L.); 2Electrophysiology Laboratory, Department of Ophthalmology, University Hospitals, Case Western Reserve University, Cleveland, OH 44106, USA; minzhong.yu@uhhospitals.org

**Keywords:** proliferative diabetic retinopathy (PDR), pan-retinal photocoagulation (PRP), conbercept, optical coherence tomography angiography (OCTA), vessel density (VD)

## Abstract

Background: This study compares the changes in retinal vessel density (VD) after pan-retinal photocoagulation (PRP) and intravitreal conbercept (IVC) treatment in proliferative diabetic retinopathy (PDR) eyes by optical coherence tomography angiography (OCTA). Methods: A total of 55 treatment-naïve PDR eyes were included in this retrospective study. Of these, 29 eyes were divided into a PRP group, and 26 eyes were divided into an IVC group based on the treatment they received. OCTA was performed to measure macular and papillary VD at each follow-up in both groups. Results: The macular VD for superficial capillary plexus (SCP), deep capillary plexus (DCP), choriocapillaris (CC) and papillary VD for radial peripapillary capillary (RPC) between the two groups demonstrated no significant difference at baseline and month 12 (*p* > 0.05). The paired *t*-test results showed that the macular VD for SCP, DCP, CC and papillary VD for the RPC at month 12 did not differ to the baseline in each group (*p* > 0.05). Conclusions: During the 12-month follow-up, there was no significant change of macular and papillary VD between the PRP and IVC treatment in PDR eyes. Additionally, compared to the baseline, there were no significant changes of macular and papillary VD after either the PRP or IVC treatment. Considering the decrease in VD as DR progress, both treatment modalities can potentially prevent macular and papillary VD loss in PDR.

## 1. Introduction

The traditional treatment of proliferative diabetic retinopathy (PDR) is pan-retinal photocoagulation (PRP) established by the Early Treatment Diabetic Retinopathy Study (ETDRS) [1]. The effectiveness of PRP for preventing severe vision loss in PDR patients has been clinically proved for decades. However, due to the retinal destruction caused by PRP, its side effects include loss of visual field, decrease in night vision and worsening diabetic macular edema (DME) [2]. More recently, a clinical trial PROTOCOL-S [3,4] found that intravitreal ranibizumab was noninferior to PRP for the management of PDR for up to 5 years. In addition, ranibizumab cohorts had lower rates of developing vision-impairing DME and less visual field loss compared to PRP cohorts [3,4]. PRP is no longer the obvious choice for the management of PDR.

Optical coherence tomography angiography (OCTA) is a new technique to obtain depth-resolved imaging of the retinal vasculature. In order to generate motion contrast OCTA images, multiple repeated B scans are performed at the same location in the retina, and structural images are derived using blood flow signal [5]. Diabetic retinopathy (DR) is characterized by retinal vascular abnormalities, such as microaneurysms, hemorrhages and neovascularization. Compared to the fluorescein angiography (FA), which is still the gold-standard test for evaluating retinal vasculature of DR, OCTA has several advantages, including the fact that it is noninvasive, shorter time for image acquisition and less affect by hemorrhage or leakage. It is pivotal that OCTA allows visualization of microvasculature and quantification of the retinal capillaries, which is very helpful for the evaluation of DR. The most commonly used application of OCTA in DR management is quantitative measures of vessel density (VD), which is defined as the proportion of blood vessel area over the total measured area. VD is associated with DR severity and can be used to predict the treatment outcome in DR patients [6]. Prior studies have reported that the VD changes after the treatment with anti-vascular endothelial growth factor (VEGF) or PRP treatment in PDR eyes with mixed results [7,8,9,10]. In this study, the retinal VD was compared between anti-VEGF treatment and PRP before the treatments and 12-month after the treatments. It will provide more insights for selecting treatment by comparing the effect on retinal VD between the above two treatments for PDR eyes.

## 2. Materials and Methods

This is a nonrandomized retrospective comparative study. The participants were recruited in the clinic of Zhongshan Ophthalmic Center in southern China between October 2016 and April 2020. The study adhered to the tenets of Helsinki and was approved by the institutional review board of Zhongshan Ophthalmic Center. Written informed consent was not obtained since the data were collected retrospectively.

### 2.1. Participants

Patients were included if they had at least one treatment-naïve PDR eye with best corrected visual acuity (BCVA) of more than 40 letters with the chart developed for use in the Early Treatment Diabetic Retinopathy Study (ETDRS). All patients underwent a comprehensive ophthalmic examination, including BCVA, slit-lamp biomicroscopy, dilated ophthalmoscopy, fundus fluorescein angiography (FFA), 7-field fundus photographs and OCTA. Patients were excluded if they had diabetic macular edema (DME), vitreous hemorrhage, media opacities that could affect image acquisition and other ocular diseases that might confound the results, such as uveitis, glaucoma, macular degeneration and high myopia. Eyes with a history of anti-VEGF treatment, PRP surgery and vitrectomy were also excluded.

### 2.2. Treatment Protocol

The patients were divided into two groups based on the treatment they received. PRP group received PRP surgery with parameters set as impulse time 100 ms, and spot size 300–500 μm with 1 spot-sized space in between. The laser power was gradually increased until a grey-white lesion on the retina was achieved and 1200–1600 total shots were delivered in 3 sessions (IRIDEX IQ 577TM laser systems from RIDEX Inc., Santa Clarita, CA, USA). Conbercept (0.5 mg/0.05 mL, Chengdu Kanhong biotech, Inc., Chengdu, China) was used for all anti-VEGF treatments. Intravitreal conbercept (IVC) group treated with 3 + PRN (pro re nata) regimen. The IVC treatment was repeated when the recurrence of a new vessel (NV) was detected after 3 initial loading shots. The PRP group was followed every 3 months for 1 year and the IVC group was followed monthly for 1 year. For both groups, focal or grid macular photocoagulation was performed if there was persistence of macular edema on follow-up visits.

### 2.3. OCTA Image Acquisition

The spectral-domain OCTA images were captured using the RTVue-XR100-2 OCTA device (Optovue Inc., Fremont, CA, USA). Split-spectrum amplitude decorrelation angiography (SSADA) algorithm was used to extract the blood flow information from 2 successive B-scan images. AngioVue provided a built-in software (version 2017.1.0.155) to segment retina and calculate VD of different section automatically. Manual segmentation was performed when scan errors were identified. The built-in software calculates VD by extracting a binary image of retinal vessels from a greyscale OCTA image and then calculating the percentage of retinal vessels pixels in the scan area. The OCTA system used motion correction technology and a projection artifacts removal (PAR) algorithm to reduce artifacts [11]. Superficial capillary plexus (SCP) was defined as from internal limiting membrane (ILM) to 10 μm above inner plexiform layer (IPL). Deep capillary plexus (DCP) was defined as from 10 μm above IPL to 10 μm below outer plexiform layer (OPL) and the choriocapillaris (CC) was defined as a section 10 to 30 μm below the retinal pigment epithelium (RPE). The whole macular angiography scan area is 3 × 3 mm. The foveal region is defined as a 1-mm ring centered on the fovea and the parafoveal region is defined as the zone between the 1-mm and 3-mm concentric rings centered on the fovea. Foveal avascular zone (FAZ) represents a region absent of capillaries at the center of fovea. The overall FAZ was measured automatically by the built-in software and manually correction was implemented if needed. We only obtained an optic disc angiography scan at the radial peripapillary capillary (RPC) slab, which extends from the ILM to the posterior boundary of the retinal nerve fiber layer (RNFL). The papillary angiography scan covers an area of 4.5 × 4.5 mm centered around the optic disc. The software automatically calculated the whole image VD, inside disc VD and peripapillary VD. The images with a scan quality better than 7 were considered eligible.

### 2.4. Statistical Analysis

Baseline demographics and variables were compared between the two groups using an unpaired *t*-test and Pearson’s chi-squared test. An unpaired *t*-test was performed to test the difference between the two groups at the baseline, month 12 and the changes from baseline to month 12. A paired *t*-test was performed to test the difference before and after treatment in each group. A two-sided *p* < 0.05 was considered to be statistically significant. Statistical analyses were performed using Prism 8 (Version 8.4.3).

## 3. Results

A total of 55 PDR eyes from 55 patients were included, in which 29 eyes were in the PRP group and 26 eyes were in the IVC group. The demographics and baseline characteristics of the two groups are shown in Table 1. There was no significant difference between the two groups in terms of age, gender, glycated hemoglobin, duration of diabetes, BCVA and central foveal thickness (CFT). All of the 55 patients were diagnosed with type 2 diabetes.

During the 12-month follow-up, the mean BCVA decreased from 74.10 ± 9.66 letters to 72.52 ± 6.45 letters in the PRP group and improved from 75.57 ± 9.94 letters to 76.83 ± 5.68 letters in the IVC group. The changes of BCVA in the PRP group (−1.58 ± 4.41 letters) and the IVC group (1.26 ± 5.57 letters) were significantly different (*p* = 0.044). The mean CFT increased from 285.8 ± 40.74 μm at the baseline to 313.33 ± 43.92 μm at month 12 in the PRP group and decreased from 296.8 ± 42.38 μm at the baseline to 268.8 ± 30.35 μm at month 12 in the IVC group. The changes of CFT in the PRP group (27.59 ± 40.68 μm) and the IVC group (−28.0 ± 27.35 μm) were significantly different (*p* < 0.001). There were 11 eyes (37.9%) that needed macular grid laser photocoagulation for DME in the PRP group and none in the IVC group (*p* < 0.001).

### Vessel Density and FAZ on OCTA

The baseline macular VD for SCP, DCP, CC and papillary VD for the RPC of the two groups were similar, which are shown in Table 2 (*p* > 0.05). At month 12, the macular and papillary VD demonstrated no significant difference between the two groups (*p* > 0.05) (Table 3). The paired *t*-test revealed that the macular VD for SCP, DCP, CC and papillary VD for the RPC at month 12 did not differ significantly to the baseline in each group (Table 4, Figure 1 and Figure 2). The FAZ of the two groups demonstrated no significant difference at the baseline and month 12 (*p* > 0.05) (Table 2 and Table 3). Additionally, no significant difference between the baseline and month 12 was found in each group (Table 4).

## 4. Discussion

The effectiveness of PRP for treating PDR was clinically confirmed decades ago but the mechanism of action of PRP remains unclear. The main theory is that PRP works by destroying the peripheral retinal pigment epithelium (RPE) and adjacent photoreceptors to reduce retinal oxygen consumption. In consequence, hypoxia of the central retina is ameliorated [12]. The improved oxygenation of the retina regulates its blood flow mainly by oxygen tension [13]. The increase in oxygen tension after PRP treatment leads to constriction of the retinal arterioles [14,15]. Wilson et al. [14] found that the mean retinal arteriolar diameter decreased by 9.7% after PRP treatment in PDR eyes. Afterwards, Mendrinos et al. [15] observed that the retinal arteriolar diameter decreased by 13.8% following PRP in severe non proliferative diabetic retinopathy (NPDR) and PDR eyes. However, the effect of PRP on the retinal capillaries has not been fully understood yet.

Increasing evidence shows that retinal VD decreases with the development of the severity of DR [16,17]. In our hypothesis, the amelioration of hypoxia in the retina after PRP treatment slows down the loss of retinal capillaries in the above situation. Several studies have investigated the changes of macular VD after PRP treatment with OCTA. Li et al. [8] found that the macular VD increased 1 day after PRP but decreased 6 months after PRP in severe NPDR eyes. Lorusso et al. [9] found that PRP did not change macular VD in PDR eyes at a 6-month follow-up. Fawzi et al. [18] reported no significant change in macular VD after PRP in PDR eyes, but the adjusted flow index (AFI, a self-created surrogate metric of blood flow) suggested that the capillaries flow was improved. Our results are in line with Lorusso’s study, where the PRP group demonstrated no significant changes of macular and papillary VD after PRP treatment.

Conbercept is a newly developed recombinant fusion protein anti-VEGF drug that has high affinity to all VEGF isoforms [19]. Conbercept was approved for the treatment of wet age-related macular degeneration, choroidal neovascularization secondary to pathologic myopia, retinopathy of prematurity and DME in China. Although the use of conbercept in PDR treatment is currently off-label, several studies have confirmed its effectiveness [20,21]. VEGF inhibition has been reported to induce retinal arteriolar constriction [22] and the vasoconstriction was a concern that might increase hypoxic damage to macular capillaries and lead to further macular ischemia [23]. However, our results revealed that macular and papillary VD did not change after the IVC treatment. Zhu et al. [7] found that macular VD for the SCP was improved after IVC treatment in eyes with DME. Conti et al. [24] and Alagorie et al. [10] also found no change of macular VD after anti-VEGF treatment. In addition, Alagorie’s study was a randomized clinical trial with retrospective analysis of the results in PDR eyes treated by intravitreal aflibercept (IVA) monthly or quarterly. Their results demonstrated that macular VD did not change significantly after 12 months of IVA treatment in both regimens. This suggests that even a monthly treatment of anti-VEGF drugs does not affect macular VD. As mentioned above, vasoconstriction of retinal arterioles was found in the early stage after anti-VEGF and PRP treatment. Theoretically, the vasoconstriction should result in a decrease in VD. However, there is no evidence suggesting the vasoconstriction is permanent; therefore, it should not affect our results of VD at month 12. If that is the case, unchanged VD at month 12 indicates that VD increased relatively compared to baseline. Regardless, there is no cause for concern that anti-VEGF treatment might exacerbate macular and peripapillary ischemia in DR eyes.

The FAZ is a round or oval avascular area with a 500–600-micrometer diameter in heathy eyes [25]. Enlargement and irregularity of the FAZ are important indications of diabetic macular ischemia [25]. A previous study has found that patients with PDR have a larger FAZ compared to normal eyes and enlargement of the FAZ indicates an increase in non-perfusion [26]. Lorusso et al. found no change of the FAZ six months after PRP treatment in PDR eyes [9]. Felipe et al. found no change of the FAZ after anti-VEGF treatment in DR eyes with OCTA [24]. Similar to VD change, our results showed that the FAZ did not change significantly after IVC and PRP treatment in a year.

In the CLARITY study [27], which compared PRP and IVA for treating PDR, the mean BCVA changes from baseline to 52 weeks were −2.9 letters and 1.3 letters in the PRP and IVA group. Additionally, the mean changes of CFT from baseline to 52 weeks were 27.5 μm and −12.2 μm in the PRP and IVA group. Our results in BCVA and CFT were quite similar to the CLARITY study. Both suggest that anti-VEGF treatment provides better visual acuity and anatomical outcomes compared to PRP in the management of PDR in 1 year. However, the purpose of our study was to compare the effects of IVC and PRP treatments on VD in PDR. In order to obtain more accurate OCTA data, many PDR patients were excluded from this study because of insufficient data and/or the poor quality of the OCTA images. Although conbercept was shown to be effective for treating PDR in our retrospective study with a selective sample, it still needs to be validated in a randomized clinical trial. Moreover, IVC treatment for PDR is significantly more expensive than PRP in China, so the cost-effectiveness of the two regimens needs to be evaluated in the future.

Reducing artifacts of OCTA image acquisition was the biggest challenge in this study. We excluded eyes with DME at baseline to avoid the impact of DME on segmentation artifacts. The PAR algorithm was used to reduce the projection of artifacts from superficial vessels to deeper layers. Nonetheless, the complete removal of projection artifacts was impossible in the current stage and the accuracy of OCTA for measuring VD was compromised as a result. Since all of the OCTA data were obtained using the same OCTA system and algorithm, a small number of projection artifacts left should not affect our results. Hopefully, this problem will be resolved in the next generation of OCTA. There are limitations that need to be noted when interpreting our results, including the retrospective design, small sample size and undertreated DME of the PRP group. The intermediate results were not shown mainly due to insufficient data but the long-term effects of the two treatments were presented in this study. In addition, we only scanned the macular and the optic disc area. A new swept-source widefield OCTA to obtain capillaries perfusion of not only posterior pole but also mid-periphery of the retina has become available recently. With a much larger scanned area of the retina and larger sample size, the difference between the retinal VD affected by the PRP and anti-VEGF treatments will be detectable in the future.

In conclusion, our results revealed that there was no significant change of macular and papillary VD after PRP or anti-VEGF treatment in a year. Additionally, no difference in the changes of VD was found between the two regimens. These findings suggest that neither anti-VEGF nor PRP reduced macular and papillary VD in the treatment of PDR. Considering the decrease in VD as DR progress, both treatment modalities can potentially prevent VD loss in PDR.

## Figures and Tables

**Figure 1 jcm-10-04484-f001:**
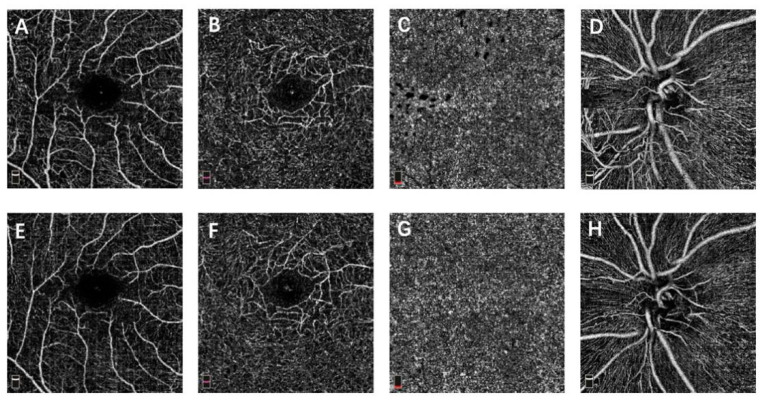
OCTA images of a PDR eye that received IVC surgery. (**A**,**E**) macular OCTA en face slab of superficial capillary plexus at baseline and month 12. (**B**,**F**) macular OCTA en face slab of deep capillary plexus at baseline and month 12. (**C**,**G**) macular OCTA en face slab of choriocapillaris at baseline and month 12. (**D**,**H**) papillary OCTA en face slab of radial peripapillary capillary at baseline and month 12.

**Figure 2 jcm-10-04484-f002:**
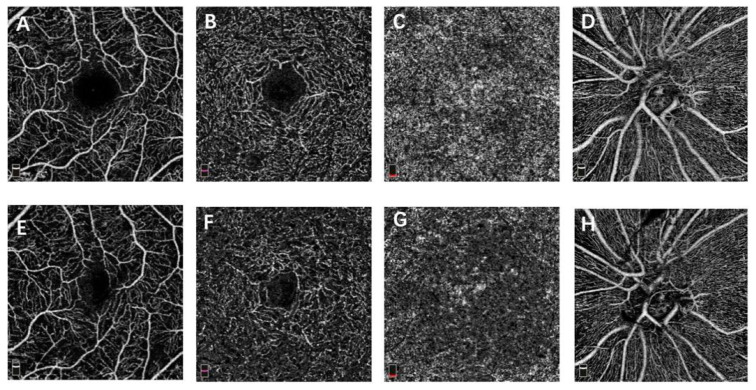
OCTA images of a PDR eye that received PRP treatment. (**A**,**E**) macular OCTA en face slab of superficial capillary plexus at baseline and month 12. (**B**,**F**) macular OCTA en face slab of deep capillary plexus at baseline and month 12. (**C**,**G**) macular OCTA en face slab of choriocapillaris at baseline and month 12. (**D**,**H**) papillary OCTA en face slab of radial peripapillary capillary at baseline and month 12.

**Table 1 jcm-10-04484-t001:** Patients’ demographics and baseline characteristics (mean ± SD).

	PRP (*n* = 29)	IVC (*n* = 26)	*p*
Age (years)	54.00 ± 10.46	55.12 ± 11.08	0.763 *
Gender (male/female)	13/16	15/11	0.341 ^†^
Glycated hemoglobin (%)	8.02 ± 1.24	7.63 ± 1.75	0.342 *
Duration of diabetes (years)	12.24 ± 6.73	10.56 ± 6.55	0.353 *
BCVA (letters)	74.10 ± 9.66	75.57 ± 9.94	0.595 *
CFT (μm)	285.8 ± 40.74	296.8 ± 42.38	0.343 *

Abbreviations: SD, standard deviation; PRP, pan-retinal photocoagulation; IVC, intravitreal conbercept; BCVA, best-corrected visual acuity; CFT, central foveal thickness. ^†^ Pearson χ^2^ test, * unpaired *t* test.

**Table 2 jcm-10-04484-t002:** Vessel density (%) and FAZ area of the two groups at baseline (mean ± SD).

Layers and Region	PRP (*n* = 29)	IVC (*n* = 26)	*p*
SCP—whole scan	30.86 ± 4.11	31.40 ± 3.56	0.621
SCP—foveal	12.65 ± 3.60	11.60 ± 3.67	0.307
SCP—parafoveal	32.14 ± 4.59	33.45 ± 3.88	0.281
DCP—whole scan	42.22 ± 5.33	43.14 ± 5.48	0.542
DCP—foveal	28.04 ± 7.19	25.43 ± 6.94	0.194
DCP—parafoveal	43.99 ± 6.40	45.75 ± 5.92	0.313
CC—whole scan	52.75 ± 4.24	51.53 ± 3.74	0.286
CC—foveal	52.73 ± 6.19	49.37 ± 6.82	0.068
CC—parafoveal	52.77 ± 4.88	53.02 ± 3.28	0.838
RPC—whole scan	44.89 ± 3.19	43.66 ± 1.87	0.108
RPC—inside disc	40.69 ± 4.37	38.64 ± 5.97	0.159
RPC—peripapilary	47.31 ± 3.21	46.39 ± 2.23	0.244
FAZ area (mm^2^)	0.30 ± 0.09	0.33 ± 0.07	0.264

Abbreviations: SD, standard deviation; PRP, pan-retinal photocoagulation; IVC, intravitreal conbercept; SCP, superficial capillary plexus; DCP, deep capillary plexus; CC, choriocapillaris; RPC, radial peripapillary capillary; FAZ, foveal avascular zone.

**Table 3 jcm-10-04484-t003:** Vessel density (%) and FAZ area of the two groups at month 12 (mean ± SD).

Layers and Region	PRP (*n* = 29)	IVC (*n* = 26)	*p*
SCP—whole scan	33.06 ± 5.48	33.04 ± 4.82	0.987
SCP—foveal	12.37 ± 2.34	11.23 ± 2.73	0.102
SCP—parafoveal	33.67 ± 6.48	34.88 ± 5.45	0.477
DCP—whole scan	42.72 ± 6.78	43.30 ± 5.39	0.743
DCP—foveal	26.91 ± 8.02	23.98 ± 5.09	0.134
DCP—parafoveal	45.48 ± 7.06	45.85 ± 5.71	0.840
CC—whole scan	53.19 ± 5.32	53.67 ± 6.45	0.772
CC—foveal	54.30 ± 6.95	51.05 ± 8.15	0.127
CC—parafoveal	53.12 ± 5.47	53.74 ± 6.56	0.709
RPC—whole scan	44.54 ± 2.56	44.36 ± 2.09	0.779
RPC—inside disc	40.62 ± 6.68	40.36 ± 9.21	0.908
RPC—peripapilary	47.20 ± 3.21	47.47 ± 2.43	0.765
FAZ area (mm^2^)	0.33 ± 0.05	0.35 ± 0.06	0.302

Abbreviations: SD, standard deviation; PRP, pan-retinal photocoagulation; IVC, intravitreal conbercept; SCP, superficial capillary plexus; DCP, deep capillary plexus; CC, choriocapillaris; RPC, radial peripapillary capillary; FAZ, foveal avascular zone.

**Table 4 jcm-10-04484-t004:** Paired *t*-test results between baseline and month 12 in PRP group and IVC group.

Layers and Region	Significance of Difference between Baseline and Month 12 (*p* Value)	
	PRP Group	IVC Group
SCP—whole scan	0.092	0.149
SCP—foveal	0.742	0.386
SCP—parafoveal	0.151	0.341
DCP—whole scan	0.713	0.913
DCP—foveal	0.522	0.303
DCP—parafoveal	0.395	0.942
CC—whole scan	0.670	0.228
CC—foveal	0.331	0.311
CC—parafoveal	0.798	0.571
RPC—whole scan	0.674	0.085
RPC—inside disc	0.964	0.744
RPC—peripapilary	0.879	0.096
FAZ area	0.602	0.464

Abbreviations: SD, standard deviation; PRP, pan-retinal photocoagulation; IVC, intravitreal conbercept; SCP, superficial capillary plexus; DCP, deep capillary plexus; CC, choriocapillaris; RPC, radial peripapillary capillary; FAZ, foveal avascular zone.

## Data Availability

The data presented in this study are available on request from the corresponding author. The data are not publicly available due to privacy restrictions.

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
