# Peer review of "Comparison of the Effect of Pan-Retinal Photocoagulation and Intravitreal Conbercept Treatment on the Change of Retinal Vessel Density Monitored by Optical Coherence Tomography Angiography in Patients with Proliferative Diabetic Retinopathy"

_jcm, 2021, doi:10.3390/jcm10194484_

Round 1

Reviewer 1 Report

The authors have shown IVC treatment for diabetic retinopathy would  not cause significant complication of retinal ischemia.  Although the study is  retrospective and may have some skew in the selection of patients and treatments, the obtained results are surprising to this reviewer.

Please describe some comments on the following points in the Discussion.

  1. Please compare PRP and IVC in the point of cost and benefit view for PDR patients.
  2. There are changes in BCVA and CFT between the treatment groups. Please comment on this.
  3. What do the authors think about other anti-VEGF medications that may cause ischemia?

Reviewer 2 Report

The Authors report an interesting comparison between PRP and Conbercept effects on VD in DME treatment. Although the study is interesting some issues need to be addressed.

Please define VD: is it calculated on a binary of skeletonized image?

Were both type 1 and 2 diabetes included? In the same percentage in each group? Were some differences noted in neovessels regression and VD changes?

What about the FAZ area?

Looking at figures 1 and 2 some projection artifacts seem to affect the DCP images. Were the artifacts and segmentation checked? This could be a limit and should be taken into account.

Lines 188-191: the mechanisms of action of PRP is still matter of discussion. The one reported is one of the supposed mechanisms.

How many patients needed rescue macular laser treatment?

What about neovessels regression? And safety issues? Vitreous hemorrhages? Systemic and local adverse events?

Lines 230-231: “The strength of this study is that we compared the effects of two mainstream regimens for treating PDR on macular and papillary VD.” Conbercept is still off label for PDR and still not worldwide used, thus this sentence should be reviewed and the current use of conbercept should be better delineated (Indications, countries where it is approved etc)

As mentioned by the Authors the effects of PRP and anti-VEGF may be different on arterioles and capillaries. This should be taken into account when discussing the results on SCP and DCP/RPCP. And the presence of artifacts may be a significant issue.

Moreover, where only baseline and 12-months follow-up analysed? Are there any intermediate result?

Round 2

Reviewer 2 Report

With regards to point 2: "Were both type 1 and 2 diabetes included? In the same percentage in each group? Were some differences noted in neovessels regression and VD changes?

Response:Patients with type 1 diabetes was not excluded in this study but none of our patients were type 1. So we do not know how PRP and IVC treatment effects on VD differ in patients with different types of diabetes. We have added this information at the baseline characteristics (please see in line 133 )."

Comment: at line 133 it is reported "All of 55 patients were diagnosed with type I diabetes." Did the Authors mean type II?

With regard to point 7: "What about neovessels regression? And safety issues? Vitreous hemorrhages? Systemic and local adverse events?

Response:The above questions are crucial for evaluating anti-VEGF and PRP treatment in PDR. In this paper, however, we want focus on the change of VD after anti-VEGF and PRP treatment. We have collected a larger number of data of PDR patients treated with PRP and IVC. But many of our patients were excluded from this study due to insufficient OCTA data. In order to report the efficacy and safety of these two treatments for PDR more clearly and objectively, we are still collecting data and preparing another paper to cover these issues."

Comment: This should be mentioned in the Manuscript

With regard to point 9:"As mentioned by the Authors the effects of PRP and anti-VEGF may be different on arterioles and capillaries. This should be taken into account when discussing the results on SCP and DCP/RPCP. And the presence of artifacts may be a significant issue.

Response:The vasoconstriction of arterioles after PRP and anti-VEGF treatment was found in the early stage after treatment. If the vasoconstriction of arterioles is permanent, but the overall VD remains the same at month 12 in our results, this might indicate that the capillary density increased relatively. This is consistent with our conclusions that these two treatments have potential protection of VD loss. So we believe the vasoconstriction of arterioles should not affect our results at 1 year. And as mentioned above, since all of OCTA data were obtained using the same machine and algorithm, a small number of artifacts shouldn’t affect our results."

The concept expressed in response to point 9 on vasoconstrction is interesting and needs to be mentioned as well as the reason for showing only 1 year results explained in the last point

Author Response

With regards to point 2: "Were both type 1 and 2 diabetes included? In the same percentage in each group? Were some differences noted in neovessels regression and VD changes?

Response:Patients with type 1 diabetes was not excluded in this study but none of our patients were type 1. So we do not know how PRP and IVC treatment effects on VD differ in patients with different types of diabetes. We have added this information at the baseline characteristics (please see in line 133 )."

Comment: at line 133 it is reported "All of 55 patients were diagnosed with type I diabetes." Did the Authors mean type II?

Response:Thank you for pointing out the mistake. It’s type II.

With regard to point 7: "What about neovessels regression? And safety issues? Vitreous hemorrhages? Systemic and local adverse events?

Response:The above questions are crucial for evaluating anti-VEGF and PRP treatment in PDR. In this paper, however, we want focus on the change of VD after anti-VEGF and PRP treatment. We have collected a larger number of data of PDR patients treated with PRP and IVC. But many of our patients were excluded from this study due to insufficient OCTA data. In order to report the efficacy and safety of these two treatments for PDR more clearly and objectively, we are still collecting data and preparing another paper to cover these issues."

Comment: This should be mentioned in the Manuscript

Response:We think efficacy and safety issues better be reported in RCT. We have mentioned this in line 263-268.

With regard to point 9:"As mentioned by the Authors the effects of PRP and anti-VEGF may be different on arterioles and capillaries. This should be taken into account when discussing the results on SCP and DCP/RPCP. And the presence of artifacts may be a significant issue.

Response:The vasoconstriction of arterioles after PRP and anti-VEGF treatment was found in the early stage after treatment. If the vasoconstriction of arterioles is permanent, but the overall VD remains the same at month 12 in our results, this might indicate that the capillary density increased relatively. This is consistent with our conclusions that these two treatments have potential protection of VD loss. So we believe the vasoconstriction of arterioles should not affect our results at 1 year. And as mentioned above, since all of OCTA data were obtained using the same machine and algorithm, a small number of artifacts shouldn’t affect our results."

The concept expressed in response to point 9 on vasoconstrction is interesting and needs to be mentioned as well as the reason for showing only 1 year results explained in the last point

Response:Thanks again for your time and suggestions. We have mentioned the above question in line 231-235 and line 280-281.
